# Effect of Immunocastration on Culled Sows—A Preliminary Study on Reproductive Tract, Carcass Traits, and Meat Quality

**DOI:** 10.3390/vetsci10100600

**Published:** 2023-10-02

**Authors:** Sofia Botelho-Fontela, Gustavo Paixão, Ricardo Pereira-Pinto, Manuela Vaz-Velho, Maria dos Anjos Pires, Rita Payan-Carreira, Luís Patarata, José Lorenzo, António Silva, Alexandra Esteves

**Affiliations:** 1Animal and Veterinary Research Centre (CECAV), AL4Animals, University of Trás-os-Montes and Alto Douro, 5000-801 Vila Real, Portugallpatarat@utad.pt (L.P.); jasilva@utad.pt (A.S.); alexe@utad.pt (A.E.); 2CISAS—Center for Research and Development in Agrifood Systems and Sustainability, Instituto Politécnico de Viana do Castelo, 4900-347 Viana do Castelo, Portugal; rpinto@ipvc.pt (R.P.-P.); mvazvelho@estg.ipvc.pt (M.V.-V.); 3CHRC—Comprehensive Health Research Centre, Department of Veterinary Medicine, University of Évora, Pole at Mitra, 7002-554 Évora, Portugal; rtpayan@uevora.pt; 4Centro Tecnológico de la Carne de Galicia 4, 32900 San Cibrao das Viñas, Spain; jmlorenzo@ceteca.net

**Keywords:** meat quality, female immunocastration, culled sows, fatty acids

## Abstract

**Simple Summary:**

This study focuses on the Bísaro breed and the need to prevent unwanted pregnancies and boar taint in cull sows of this breed. Three immunocastration protocols using Improvac^®^ were tested and compared to a control group under the same conditions. The immunocastrated sows had significantly lighter reproductive tracts compared to the intact sows, indicating a regression in reproductive activity due to GnRH immunization. However, there were no significant differences in meat quality traits. Therefore, immunocastration can effectively suppress ovarian activity in culling sows without negatively impacting pork quality. This research offers a practical solution to manage reproductive issues, ensuring both animal welfare and meat quality.

**Abstract:**

The Bísaro pig is a Portuguese autochthonous breed greatly appreciated for its meat quality and is mainly reared outdoors. Immunocastration could be a solution to avoid undesirable pregnancies and boar taint in cull sows. The present study tested three immunocastration protocols (with Improvac^®^) according to their reproductive cycle. The first inoculation was performed two weeks after farrowing (IM1, n = 5), at the beginning of estrus (IM2, n = 5), and one week after the end of estrus (IM3, n = 5), followed by a second administration four weeks apart. A control group (C, n = 5) was also included in the same housing conditions. The sample collection included the reproductive tract for morphometric evaluation, neck fat for the quantification of boar taint compounds, and a portion of the Longissimus thoracis et lumborum for meat quality trait assessment. The reproductive tracts from intact sows (C) were significantly heavier compared to the immunocastrated groups (*p* < 0.05) (1.403 kg C to 0.508 kg IM1, 0.590 kg IM2, and 0.599 kg IM3), suggesting the regression of the reproductive tract to nonstimulated conditions due to immunization against GnRH. The IM1 group exhibited significantly smaller reproductive tract measurements compared to group C for most of the evaluated segments (*p* < 0.05). No marked differences were observed in the meat quality traits. Therefore, immunocastration can be used in culling sows to avoid ovarian activity, and it is not detrimental to pork quality traits.

## 1. Introduction

Females at the end of their productive life or with unfit performance are culled from the farm and sent to slaughter. This is a crucial decision for the farm’s efficiency, as it directly affects its economic sustainability [1]. The most frequent reasons for culling include reproductive failure, lameness, aging, and reduced health, which affect the sow’s reproductivity and longevity [2,3]. Most sows are culled immediately after weaning their last litter [4], when their physical condition is at its worst [5,6,7,8]. Even though most sows gain fat during pregnancy, they lose that reserve during lactation [7], as there is high energy demand. Lactation triggers body condition loss, along with the absence of cyclic ovarian activity [9,10]. The loss of body condition is even more pronounced when prolonged lactations are used, as is usual in the Bísaro production system [11].

Both sow carcasses and meat are poorly valued in retail cuts due to their undesirable aromas and flavors [12], sometimes addressed as ‘sow taint’ [13]. As such, this meat is mostly used in processed meat products, such as fresh pork sausages and meatballs [1,12,14]. The causes of boar taint in sows have not been identified, and there are no studies to date on the origin and composition of these atypical aromas and flavors in sow meat. In contrast, boar taint has been widely investigated. Boar taint is an objectionable odor often perceived when cooking meat from entire male pigs [15,16], and its origin has been associated with androstenone and skatole. Androstenone is a testicular steroid produced in Leydig cells with a urine-like smell. It is reduced to α-androstenol and β-androstenol, which are excreted in saliva, acting as pheromones with a role in the pigs’ reproductive function [17,18]. Skatole is a tryptophan breakdown product that exhibits an intense fecal odor [15]. Even though, in theory, androstenone should only be found in males, there are reports that it is also found in females [19]. Skatole values are higher in entire males, but they can also be found in smaller amounts in barrows and gilts [20] and present as the main taint contributor when the slaughter weights are low [21].

Castration is the most effective method to avoid undesirable meat aromas [21]. In China, autochthonous breeds are surgically castrated, both males and females, to avoid sexual odor and growth delays [21]. Although surgical castration is still used in male Bísaro pigs in the European Union [22], even in culling males using anesthesia and analgesia, the surgical castration of female pigs is rare due to the difficulty of the associated procedure. Therefore, fattening females or culling sows are usually intact.

A sustainable alternative to surgical castration is immunization against gonadotropin-releasing hormone (GnRH), a key regulatory hormone orchestrating reproductive function in both males and females [21]. Immunization against GnRH drives the synthesis of antibodies against this molecule, reducing its time in circulation, thus causing the involution of the reproductive system [23,24]. The suppression of ovarian cyclicity using immunocastration in female pigs has been thoroughly established [21,24,25,26,27,28].

The Bísaro pig is a Portuguese autochthonous breed greatly appreciated for its meat quality [29]. In its traditional production system, housing is characterized by a small shed with outdoor pens (52.6% of pig farms) or free-range camping systems (40.4%). On average, sows have two litters per year with a mean of nine born piglets, from which seven are weaned [30]. Producers aim to obtain approximately 8 litters, with a farm life of approximately 5 years old, but many are culled younger, even though some sows can reach 13 litters (8 years old), depending on their performance [30]. Nonpharmacological heat synchronization is implemented to meet seasonal demands for specific meat products, such as piglets for Easter and summer and dry-cured meat products for winter [11]. Most pregnant and dry sows have access to outdoor areas part or full-time for grazing; a third of Bísaro farms have farrowing crates, while another third have outdoor cabins for breeding purposes [11]. Despite leading to heavier and fatter carcasses, outdoor rearing systems improve sow nutrition (grazing behaviors) and make physical activity possible, which influences carcass and muscle traits, and therefore, meat quality [31].

Although Bísaro sows are usually culled after weaning their last litter, they are still allowed a waiting period to regain some body condition before slaughter, usually in outdoor pens. This can lead to unwanted pregnancies due to the occasional presence of domestic or wild boars [32]. Consequently, immunocastration is used as a solution, not only for the boar taint but also for the prevention of estrus and ovulation [32,33], contributing to a faster recovery of body condition.

Despite the limited literature, investigating cull sow meat characteristics is essential to improve the use of their products and gather consumers’ perceptions, leading to a possible increase in their economic value [14]. As Bísaro farms mostly rely on breeding females, increasing their economic value is paramount for their profitability and sustainability.

This study aimed to investigate the effects of Improvac^®^ (Zoetis) starting at different moments (during lactation, in estrus, and in diestrus) in culled sows on the morphometry of the female reproductive tract, the skatole and androstenone levels, the carcass traits and meat quality, and on meat sensory evaluation.

## 2. Materials and Methods

This study was conducted at a Bísaro commercial pig farm in northern Portugal (41°47′47.9″ N; 6°41′22.8″ W) between November 2019 and September 2020. The farm has approximately 60 breeding sows, as well as the nursery and growing-finishing phases. All the animals were handled according to the national [34] and European [35] regulations on animal welfare in experimental research. 

### 2.1. Study Design

#### 2.1.1. Animals

A convenience population of twenty multiparous female Bísaro pigs were used in this study. Farm management dictates that pregnant sows are kept in outdoor pens and then moved to indoor maternities for farrowing and nursing. After weaning, sows are kept in outdoor pens in groups of five animals, either to be inseminated again or to regain body condition before slaughter. During lactation, the sows were fed a commercial diet with 16.4% crude protein, 3.8% crude fat, 6.59% crude fiber, 5.35% crude ash, 0.99% calcium, 0.52% total phosphorus, 0.2% sodium, 1.00% lysine, and 0.32% methionine. Weaned and cyclic females were fed a finishing commercial diet with 11.4% crude protein, 8.15% crude fat, 6.00% crude fiber, 4.20% crude ash, 0.53% calcium, 0.47% total phosphorus, 0.4% sodium, 0.56% lysine, and 0.19% methionine. When the sows were allocated outdoors, supplementation with beets was also performed. Feed intake was ad libitum, and water was freely accessible throughout the trials. The selection of sows was performed per farm availability; the females were culled due to infertility, increased parity, increased piglet mortality rate associated with sow behavior, or other non-reproductive health conditions, taking into consideration their reproductive period. The selected animals were aged between 2 and 5 years old and had 2–6 parities recorded. On the farm, suckling is 35 days in length.

#### 2.1.2. Immunization Treatments

Each experimental protocol lasted for 8.5 weeks. The sows were vaccinated by a veterinarian against GnRH with Improvac^®^ (Zoetis, Louvain-la-Neuve, Belgium), with two injections (2 mL subcutaneous, behind the ear) four weeks apart, according to the manufacturer’s recommendations. To evaluate the most effective protocol (defined as the suppression of reproductive activity for a period of 4.5 weeks), the animals were grouped following their reproductive cycle in three protocols (Figure 1) according to the moment of the first inoculation.

-Group 1: Inoculation in middle lactation (IM1; n = 5), two weeks after farrowing (lactational anestrus), when the endogenous GnRH patterns are low [36]; the females in this group were kept indoors in individual farrowing pens with the piglets until the end of farrowing (3 weeks after the first inoculation), when they were transferred outdoors.-Group 2: Inoculation at the beginning of estrus (IM2; n = 5), when the frequency of GnRH pulses increases; [34]; the first inoculation was performed the day cyclic sows showed signs of standing heat (standing reflex); the sows in this group were allocated to outdoor pens.-Group 3: Inoculation in diestrus, one week after the end of estrus (IM3; n = 5), when the pulsatile discharges of GnRH are decreased [37]; the sows in this group were also allocated to outdoor pens.

The second inoculation was performed four weeks after the first in all the groups. The slaughter was scheduled for 4.5 weeks after the second inoculation.

A control nontreated group was also included (C; n = 5), in which the sows were also kept outdoors in similar conditions. Both the C and IM2 trials evolved in the winter, IM1 in the spring, and IM3 in the summer. 

### 2.2. Sample Collection

The animals were fasted for 12 h, and their live weight was recorded the day before slaughter. The sows were transported for a period not exceeding 2 h and sacrificed in an official slaughterhouse, complying with the current welfare hygienic legislation for food of animal origin [38,39,40].

The animals were followed along the slaughter line until evisceration to individually collect the reproductive tract and a portion of fat from the neck area into identified containers to guarantee sample traceability, then stored at 4 °C, and transported to the laboratory for further analysis. The carcasses were split lengthwise, and the hot carcass weights were individually recorded to calculate the carcass yield. The *Longissimus thoracis et lumborum* (LTL) muscle from the left side of the carcasses was used for meat quality assays. The reproductive tract was collected in a block, from the vulva to the ovaries.

### 2.3. Reproductive Tract Morphometry

At the Laboratory of Histology and Anatomical Pathology (LHAP) at the University of Trás-os-Montes and Alto Douro (UTAD), Vila Real (Portugal), the reproductive tracts were subject to visual inspection, and any macroscopic abnormality was documented. Before their overall weight was recorded, the broad ligament was removed. The vagina, cervix, uterine corpus, uterine horns, and ovaries were identified and measured according to Pires and Payan-Carreira [41], in diameter (the genital tract intact), and then in length (after the longitudinal section of the genital tract). A longitudinal incision was performed from the vagina to the bifurcation of the uterine horns to enable the identification of the anatomical segments and an internal examination of the female tract and its contents. Figure 2 depicts the anatomical limits used for each measure. For precise measurements, standard rulers were used. For the ovaries, measurements of the length, width, and depth were taken separately. The ovarian volume was estimated through the formula [ovarian length × width × depth × 0.523] as per [42]. 

The gonadosomatic index (GSI) was calculated by applying the formula [ovarian weight/bodyweight at slaughter] × 100. This provides insight into ovarian development and reproductive potential [43].

### 2.4. Physicochemical Analysis

In the slaughterhouse line, at 45 min postmortem, the pH_45min_ was recorded using a pH meter WTW 330i (Weilheim, Germany) after calibration with buffers of pH 4.01 and 7.00. The measurement was obtained in duplicate, and the probe was inserted between the 13th and 14th thoracic vertebrae. At 24 h after slaughter, the LTL muscle, between the 7th thoracic vertebrae and 3rd lumbar vertebrae, was excised, while the carcass’s left half was dissected into commercial cuts. The LTL samples were refrigerated at 4 °C and taken to Laboratório de Tecnologia, Qualidade e Segurança Alimentar (TeQSA) at the University of Trás-os-Montes and Alto Douro (UTAD), Vila Real (Portugal), trimmed of fat and connective tissue, and sliced for further analysis. The meat quality traits in the fresh cuts included the pH_24h_, color coordinates (*L**, luminosity; *a**, red-green; *b**, yellow-blue; C*, chroma; and h°, hue angle), drip loss, cooking loss, and shear force. Additionally, a sample of approximately 100 g was vacuum-packed and stored at −20 °C for subsequent chemical analyses, and another sample of approximately 400 g was vacuum-packed and stored at −20 °C for sensory evaluation. A sample of fat from the neck area was also preserved at −20 °C for further boar taint compound quantification (androstenone and skatole).

The ultimate pH (pH_24h_) was measured at the laboratory 24 h postmortem in duplicate with the same equipment as that used for the pH_45min_.

The color was measured in a slice approximately 2 cm thick with a Minolta Chroma Meter CR-310 colorimeter (Osaka, Japan) and assessed using the color coordinates *L**, *a**, and *b**, C* and h° [44]. The color was measured on the meat surface after 60 min of blooming by placing the samples in trays covered with polyethylene film and stored at 4 °C. The colorimeter was calibrated with a standard white ceramic plate, a D65 illuminant observer angle of 0°, and an aperture size of 5.0 mm.

Heme pigments were obtained by stirring 10.0 g of minced meat in 40 mL of acetone, 2 mL of water, and 1 mL of HCl 12 M for 30 s. The suspension in the sealed flask was kept for 1 h in the dark and filtered (Whatman no. 1), and the absorbance was read at 640 nm (Jasco V-530 UV/VIS Spectrophotometer, Tokyo, Japan) in a 1 cm path length cell. Absorbance values were multiplied by 680 [45] to express data in total heme pigments (mg/g) multiplied by 0.026 [46].

The drip loss was determined using the suspension method of Honikel [47]. Loin samples of approximately 65 g were put into a plastic net and suspended in a closed plastic bag for 3 days at 4 °C. The drip loss was expressed as a percentage of the mass loss relative to the initial mass of the sample.

The cooking loss was evaluated in slices of approximately 100 g with similar geometry. The samples were placed individually inside polyethylene bags and heated in a water bath at 80 °C until an internal temperature of 75 °C was reached (monitored with thermocouples introduced to the core). The heated samples were cooled in an ice bath until reaching 4 °C. The cooled samples were removed from the bag, dried with filter paper, and weighed. The cooking loss was expressed as the percentage of mass loss relative to the initial sample mass [48]. After cooking loss determination, the samples were packed in plastic bags and stored overnight at 4 °C for shear force measurements.

The meat samples used to determine the cooking losses were cut into cuboid subsamples (4 to 6) with a 1 cm^2^ cross-section and 3–4 cm in length, with the muscle fibers parallel to the length of the cuboid. After room temperature equilibrium, the sub-samples were placed with the muscle fibers perpendicular to the direction of a Warner–Bratzler rectangular hole probe coupled to a TA.XT.plus texturometer (Stable Micro Systems, Godalming, UK), with a 30 kg load cell, blade velocity of 200 mm/min, and trigger force of 5 g. The maximum shear force values were recorded, and the values were expressed in N/cm^2^.

The approximate chemical composition comprised the determination of the moisture, fat, protein, and ashes. The moisture content was determined by drying 5.0 g of the homogenized sample in an oven at 103 °C to constant mass according to ISO 1442:1997 [49]. The results were expressed as percentages by mass. The lipid content was determined following the protocol established by the American Oil Chemists’ Society (AOCS) Official Procedure Am 5-04 using the fat extractor Ankom XT10 (ANKOM Technology Corp., Macedon, NY, USA). The total nitrogen content was determined by the Kjeldahl method [50]. The digestion of 1.0 g of the homogenized sample was carried out in an Inkjel 1255P digester (beher Labor-Technik, Düsseldorf, Germany) with the addition of 1.5 catalyst pellets (7.5 g), composed of potassium sulfate and copper sulfate (Merck 1.15348) and 17 mL of concentrated sulfuric acid. The total digestion time was 150 min, with a gradual increase in temperature. For distillation, 100 mL of 35% sodium hydroxide, 50 mL of distilled water, and 30 mL of 4% boric acid with indicator (bromocresol green and methyl red) were used and subsequently titrated with 0.1 N HCl. Distillation and titration were performed on a VELP UDK 159 apparatus (Velp Scientifica Srl, Usmate, Italy). The total nitrogen content was converted to the protein content using the factor 6.25 and expressed as a percentage. The ash content was determined after the incineration of 1.50 g of the homogenized sample in a muffle furnace at 550 °C, according to ISO 936:1998 [51]. The results were expressed as a percentage by mass.

The androstenone (AND) and skatole (SKA) contents in the pig fat were measured using HPLC analysis as described by Hansen-Moller [52]. Liquid fat was extracted from the adipose tissue by solid-liquid separation after microwave heating (800 W, 2 min). Methanol was added to 1.0 g of fat, sonicated, and centrifuged. The supernatant was collected, filtered through a 0.2 µL filter, and derivatized before injection. Manual derivatization was performed at room temperature for 5 min, adding 40 µL of BF3, 50 µL of deionized water, and 75 µL of dansylhidrazine 0.1%. The HPLC system (Thermo Scientific UltiMate 3000, Waltham, MA, USA) was equipped with an AkzoNobel Kromasil 100-5C18 250 × 4.6 mm 5 µm column (Bohus, Sweden) operating at 40 °C. The composition of the mobile phase buffers was as follows: (A) acetic acid 0.1%, (B) acetonitrile, (C) tetrahydrofuran, and (D) methanol 95%. The gradient profile used, with a 2 mL/min flow, was as follows: 0.0–5.0 min: 45% A, 55% B; 5.0–6.0 min: 40% A, 55% B, 5% C; 6.0–6.1 min: 20% A, 30% B, 30% C, 20% D; 6.1–12.0 min: 40% B, 40% C, 20% D; 12.0–12.1 min: 45% A, 55% B; 12.1–13.0 min: 45% A, 55% B. Analytes were detected with a fluorescence detector with excitation at 285 nm and emission at 340 nm (0–6.0 min) for skatole and excitation at 346 nm and emission at 521 nm (6.1–13 min) for androstenone. Twenty microliters of the sample was injected. An external calibration method was used for quantification, with a calibration curve linearity coefficient of 0.999 for both compounds. The limits of detection (LoD) were 1.53 and 16.02 ng/mL for skatole and androstenone, respectively. The recovery values were 102.84% for androstenone and 99.72% for skatole. Method validation was performed, with repeatability <2.46% RSD for SKA and <6.85% RSD for AND; the intermediate precision was <2.87% RSD for SKA and <6.98% RSD for AND [53].

### 2.5. Sensory Evaluation

Twenty-two adults (thirteen females) were tested on their sensory performance based on the ISO 8586:2012 [54] methodology within 5 sessions. Six panelists were excluded due to their low performance, particularly their inability to detect either AND or SKA. The training of the remaining 16 panelists was performed according to a procedure adapted from Garrido et al. [55]. Standard solutions were prepared for AND (5α-androst-16-en-3-one, M 272.43 g/mol, Sigma-Aldrich A8008, Saint Louis, MO, USA) and SKA (3-methylindole, M 131.17 g/mol, Sigma-Aldrich M51458, Saint Louis, MO, USA) using Vaseline oil. The working solutions and the respective dilutions were stored in amber glass vials with cotton inside. Several tests were made using the vials to train the panelists, including descriptive, ranking, classification, and triangular tests, with the concentrations described in Garrido et al. [55] over the course of 16 sessions. The training was also carried out with the loins from pigs with known levels (low and high) of boar taint compounds over 5 sessions.

The meat samples were thawed at 4 °C for 24 h before assessment. Triangle tests [56,57] were used to determine if the panelists could differentiate between the control and immunocastrated sows. Samples from all animals were used in the sensory analysis tests. Every test was performed five times for each immunocastration protocol. Loin fillets (1.5 cm) were cooked and served as described in Silva [58]. Pieces measuring approximately 2 × 2 cm were extracted from the cooked fillets. No additional seasoning was included to ensure that the boar taint remained unaltered [55,59]. Two pieces of the meat samples were then carefully wrapped in aluminum foil individually and maintained at a temperature of 60 °C until they were evaluated, which took place within 30 min after the cooking process. Each panelist was served three samples in aluminum foil, two from the control group and one from an immunocastrated group and asked to identify the sample that differed from the other two. Each sample was presented with a unique three-digit identifier, and the order in which the samples were presented was randomized to mitigate any potential sequencing bias. All tests were performed in a sensory analysis laboratory with individual booths, consistent illumination conditions, and room temperature between 18 and 25 °C. Spring water at room temperature and bread were used as palate cleansers between the sample evaluations.

### 2.6. Fatty Acid Profile

Regarding the determination of fatty acids, we followed the method outlined by Domínguez et al. [60] for fat extraction and transesterification. The separation and quantification of fatty acid methyl esters (FAMEs) were carried out using a gas chromatograph (Agilent DB-23; Agilent Technologies, Santa Clara, CA, USA) equipped with a flame ionization detector (FID) and a PAL RTC-120 autosampler with a liquid injection tool (Pal System). The chromatographic conditions described by Domínguez et al. [52] were followed. To separate the FAMEs, we utilized a DB-23 fused silica capillary column (60 m, 0.25 mm i.d., 0.25 µm film thickness; Agilent Technologies, Santa Clara, CA, USA). Additionally, the n-6/n-3 and PUFA/SFA ratios were determined [61]. The results were expressed as grams per 100 g of fat.

### 2.7. Statistical Analysis

Statistical analysis was conducted with IBM SPSS Statistics for Windows (v.29) software (Armonk, NY, USA: IBM Corp). The variables were tested to assess their distribution and the normality of the data using the Shapiro-Wilk test. The data did not satisfy the assumptions of normality, and transformations were ineffective. Therefore, nonparametric tests were performed, and differences among the treatments were identified with the Kruskal-Wallis test. For triangle sensorial analysis, the data were analyzed by comparisons of the number of correct answers—the panelist considered the sample of an immunocastrated group different in the set of three—to the table of binomial law. The values expressed in the results are the means to improve understanding. Significance was declared at *p* < 0.05.

## 3. Results

The recorded reasons for culling were low prolificity, infertility, and other health-related issues. The demographic information for the studied animals is presented in Table 1. There were no differences among the groups in age and parity (*p* ˃ 0.05). No deaths were recorded in this study.

The results from the morphometric studies are shown in Table 2. The immunocastration triggered a reduction in the size and weight of the reproductive tract in most treated groups compared to the intact females.

Sows from the control group (C) presented significantly heavier genital tracts and ovaries than the immunocastrated group IM1 (*p* < 0.05). Although the weight of the genital tract of C was not significantly different between IM2 and IM3, numerically, the IM2 and IM3 reproductive tracts were less than half as heavy as those in C. The same was observed in the ovaries, as IM2 and IM3 had ovaries with less than half of the weight of those in C, despite no significant differences. The ovaries’ sizes presented no significant differences among the immunocastrated protocols. IM1 and IM2 were significantly different from C, and although, numerically, the size values of IM3 were lower than C, there were no significant differences recorded. Regarding the different immunocastration protocols, no marked morphometric differences were recorded among them. Only the cervix–vagina segment had significant differences among the immunocastrated groups, in which IM2 had longer segments than the other two protocols. The length of the uterine corpus was no different in the immunocastrated groups compared to the control group, although the uterine horns were approximately half the diameter (*p* < 0.05) of those of the C group. The gonadosomatic index (GSI) was calculated to assess the correlation between ovarian weight and live weight. The GSI was significantly lower in IM1 than in C (*p* < 0.05), but no differences were observed among the immunocastration protocols. 

The results of the studied carcass traits and physicochemical and boar taint compound quantification are presented in Table 3.

Immunocastration had no effect (*p* > 0.05) on the live or carcass weights and, consequently, did not affect the carcass yield. The meat quality physicochemical traits were not influenced by the studied immunocastration protocols (*p* > 0.05), except for the moisture content. The moisture content was significantly lower in C and IM2 (*p* ˂ 0.05). Regarding the quantification of boar taint compounds, very low amounts of skatole and androstenone were found in the neck fat, with no statistically significant differences among the groups under analysis. The sensory evaluation of the meat using triangle tests provided no differences between the C group and any of the immunocastration protocols studied (*p* > 0.05), validating the results of the meat quality traits assessed. 

The impact of immunocastration on the fatty acid profile of LTL muscle is shown in Table 4.

Total and individual saturated and monounsaturated acids did not differ significantly among the groups. On average, the total polyunsaturated fatty acid (PUFA) content was higher in the immunocastrated sows than in entire sows (*p* > 0.05). This was mainly justified by an increase in the linoleic acid (C18:2 n-6) quantity in these groups. For individual polyunsaturated fatty acids, a significant difference was found only for docosaheptaenoic acid (C22:6 n-3), which was significantly higher in the IM2 group, while IM3 presented its lowest value. For the fatty acid ratios, only n-6/n-3 showed significant differences (*p* < 0.05), with higher values in the IM3 group, whereas the control and IM2 groups registered the lowest proportion. In sum, we did not find any consistent differences in the fatty acid profile between the entire and immunocastrated sows. Regarding the treatment groups, sows that started the protocol at the beginning of estrous (IM2) seemed to have the highest n-3 content, mainly due to the higher docosaheptaenoic acid (C22:6 n-3) proportion and the lowest amount of SFA (*p* > 0.05).

## 4. Discussion

Immunocastrated gilts present lighter genital tracts than entire females [21,25,26,27,28,29,43,62,63], which is in accordance with our findings. The morphometric results observed suggest that immunocastration can be efficiently performed in all the studied phases. To our knowledge, the studies done thus far have used gilts [21,25,26,27,28,29,43,62,63]. In all the mentioned studies, prepubertal immunocastration protocols were implemented, with the second inoculation applied before puberty, i.e., before the stimulation of the genital tract by the sex steroids [28]. In the present study, we used multiparous females with fully functional reproductive tracts. Immunocastration triggered a reduction in the size and weight of the reproductive tract, associated with a decrease in sexual steroid stimulation. The marked reduction in ovarian weight (close to three times smaller in the treated groups than in the control group) was associated with a reduction in the ovarian activity induced by immunocastration. Our findings on the length of uterine horns disagree with previous studies reporting a significant reduction in their length [26,28,63], whereas, in the present study, no significant difference was found in their length despite the diameter of the uterine horns being half that of the control group. It should also be noted that the differences might have resulted from the fact that the sows in the current study were multiparous, with a parity ranging from 2 to 6. Multiple parities could contribute to an increase in the uterine dimensions, which would exacerbate individual variations that might exist. The differences observed in the GSI suggest that the reproductive tract had involuted after parturition and was not further stimulated by sexual steroids, as in the IM1 group, and the resumption of ovarian cyclicity was impaired by immunocontraceptive treatment.

Regarding the meat quality traits, no differences were found in the live or carcass weights, nor in the carcass yield, which agrees with several studies, granting that the immunocastration was performed in gilts instead of culled sows [21,24,25,26,27,64,65,66]. Studies in gilts also found no differences in the pH_45min_ and pH_24h_ (*p* > 0.05) [21,24,66,67,68,69,70]. These parameters are indicators of the extent of glycolysis in the early postmortem period and the completion of glycolysis, respectively, and are directly related to the meat quality. Pale, soft, exudative (PSE) and dark, firm, dry (DFD) meats have depreciative qualities. pH45min values below 5.8, due to a high acidification rate, can originate in PSE meats [71,72]. Neither of these cases was observed in the present study, which means that no PSE and DFD meats were found. Studies on surgical castration [21,73,74,75] and immunocastration protocols in gilts [21,24,27,65,66,67,70] observed that immunocastration does not have an influence on meat color and heme content. However, Daza et al. [64] found higher values of the *a** and C* color coordinates in immunocastrated gilts compared to entire gilts. Although most studies are in concordance with the findings in the present study on drip loss, cooking loss, and shear force [21,24,27,70], Van den Broeke et al. [67] reported that entire females had higher shear force values than those subjected to immunocastration. The moisture content presented differences in the present study, which could be explained by the intramuscular fat content. The moisture content was significantly lower in C and IM2 (*p* ˂ 0.05). A high intramuscular fat content is related to a low moisture content [71,72]. However, the means do not provide a good explanation for our results, as large variability exists in the results obtained in this study. This variability can be explained by the low number of animals studied, the intrinsic variability in culled sows (e.g., different ages and the number of parities), and even the genotypic variability characteristic of the Bísaro breed [76]. It was found that immunocastration had no effect on the chemical composition of pork [24,27,64,65,66,70]. However, Van den Broeke et al. [67] found that meat from immunocastrated gilts had higher levels of intramuscular fat compared to entire females, in contrast with Xue et al. [21], who reported that entire females had higher values of intramuscular fat than immunocastrated females.

Although no statistical differences were found regarding the boar taint compounds, their values were in the ranges between 3.51 and 7.25 ng/g for skatole and 4.29 and 12.02 ng/g for androstenone, well below the sensory threshold commonly found in the literature (200 ng/g for skatole and 500 ng/g for androstenone) [77]) and below those detected in intact Bísaro boars (31.28 ng/g on average for skatole and 244.98 ng/g on average for androstenone; data not shown). While skatole levels are typically elevated in intact males, they are also detectable in barrows and gilts, albeit in smaller quantities [20]. Furthermore, the absorption of skatole can occur through the skin via contact with feces or through the respiratory system via inhalation, which may vary depending on environmental conditions and housing facilities [78]. Regarding androstenone, the results were quite heterogeneous, and the presence of androstenone was not detected in some animals, regardless of the group. However, the presence of this pheromone is not expected when analyzing sow samples. It seems that the occurrence of androstenone is not limited solely to intact males. Although androstenone is expected to be present only in males according to theory, there are accounts indicating its presence in females [19,79,80]. This phenomenon could be attributed to the conversion of progesterone into androstadienone, followed by its subsequent transformation into androstenone [81]. Generally, female pigs typically exhibit significantly lower levels than the critical values and are unlikely to produce any sexual odor. However, it is important to consider the variations in the concentrations of 16-androstene steroids and skatole. Furthermore, it should be noted that both female pigs and castrated pigs can occasionally produce sexual, atypical odors [82].

Few studies have been performed to assess the sensory evaluation of pork from immunocastrated females, as the castration of females is not a very common practice, and mostly carried out in specific breeds [83]. Martinez-Macipe et al. [70] found that panelists only detected differences in 2 out of the 16 evaluated attributes and reported that surgically castrated females had a higher overall intensity flavor compared to entire and immunocastrated females, as well as higher sweetness compared to entire females.

Despite not being statistically significant, the mean total saturated fatty acid (SFA) proportion was greater in immuno-castrated sows than in entire sows, mostly due to higher values of palmitic acid (C16:0) and stearic acid (C18:0). These differences could have been greater if a longer interval between the second injection and slaughter had been used. Other authors also attained higher proportions of SFA in immunocastrated gilts compared to entire gilts [84,85].

The PUFA content in the present study was higher in the immunocastrated sows than in the entire sows, mainly justified by an increase in linoleic acid (C18:2 n-6). Opposite results have been reported in male [86] and female pigs [86], where entire animals had the highest content of linoleic acid (C18:2 n-6). Notwithstanding, immunocastrated pigs tend to have higher linoleic acid (C18:2 n-6) than surgically castrated animals [86,87]. From the perspective of human health and wellness, it is preferable to have a lower n-6/n-3 ratio to lower the overall diet ratio to 1:1 to 2:1 [88], along with a PUFA/SFA ratio above 0.4 [89]. In our study, all groups registered a lower mean value (0.34–0.39). Overall, there were no consistent differences in the fatty acid profile between the entire and immunocastrated sows. Similarly, neither Daza et al. [64] nor Gamero-Negrón et al. [66] found any substantial differences in the fatty acid profile between entire and immunocastrated gilts. These results might be impaired by the brief period these animals had from the second Improvac^®^ injection to slaughter and by the small sample size, although it coped with the interval to slaughter recommended by the Improvac^®^ supplier. The limited number of animals used in the current study may be viewed as a constraint; nevertheless, this study reflects actual in-farm conditions, where animals are not as easily accessible as they are in controlled research settings. Additionally, a significant challenge stemmed from our limited understanding of the underlying causes of boar taint in sows. There is a scarcity of literature on the compounds responsible for boar taint in sows, which could account for the absence of conclusive results regarding the investigated boar taint compounds.

## 5. Conclusions

To our knowledge, no immunocastration protocols have been performed in culling sows. The present study observed a marked effect on the reproductive tract, leading to the conclusion that the immunocastration was successful. The meat quality traits did not differ from the studied immunocastration protocols tested and the entire females. While no differences were observed in the testing of boar taint compounds, immunocastration, despite its higher cost, could serve as a valuable alternative during the period after weaning. This approach promotes the overcoming of the negative effects of the exhibition of heat on the recovery of body condition lost during lactation, allowing one to eventually reduce this time. The high variability expected from culled sows, as well as from Bísaro pigs, explains some of the data incongruencies, and further studies with a larger number of animals should be performed. In conclusion, immunocastration is a safe way to keep culled sows on the farm to recuperate their body condition without the risk of unwanted pregnancies, and it is not detrimental to meat quality.

## Figures and Tables

**Figure 1 vetsci-10-00600-f001:**
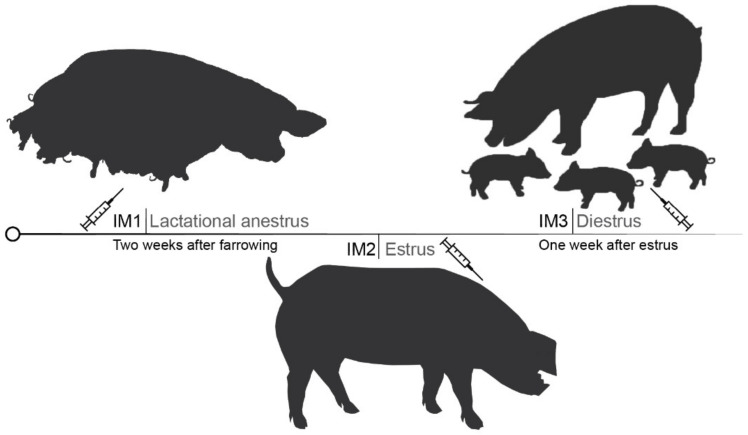
Representation of the first inoculation moment in the studied immunocastration protocol.

**Figure 2 vetsci-10-00600-f002:**
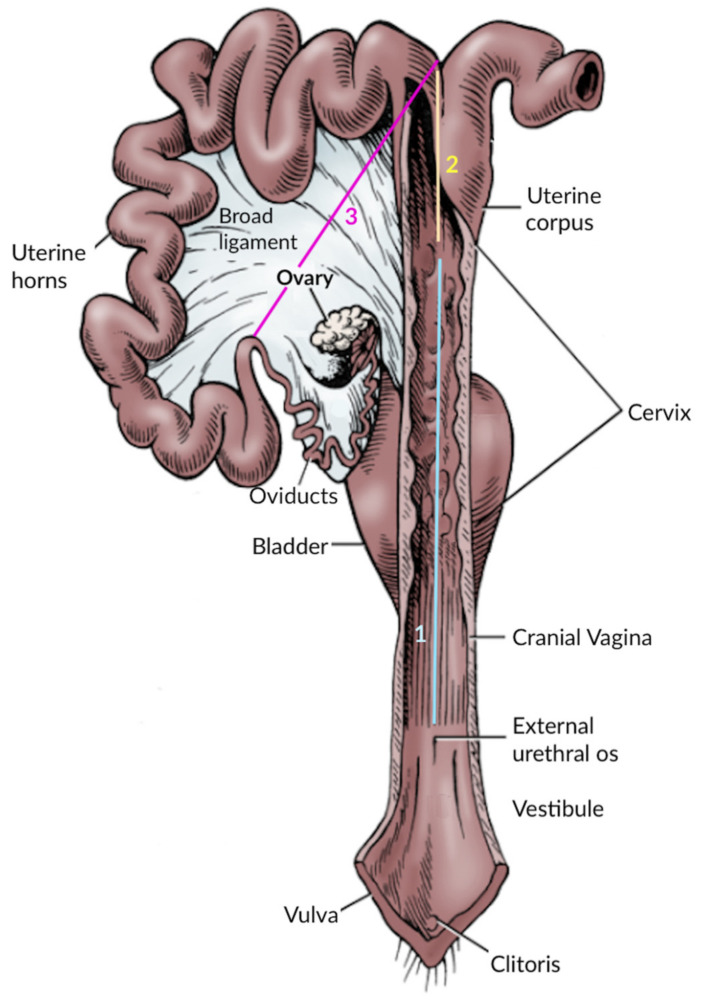
Schematic representation of a sow reproductive tract displaying the anatomical limits used to collect each segment measures: (1) the cervix–vagina measurement was collected from the beginning of the cranial vagina until the last cervical tubercule (closer to the uterus). (2) The measurement of the uterine corpus segment was collected between the last cervical tubercule to the uterine horns’ bifurcation. (3) The measures of the uterine horns segment were collected from the external bifurcation of the uterine horns to the apex of the uterine horns.

**Table 1 vetsci-10-00600-t001:** Demographic information.

Groups	Age	Parity
	Median (Q1–Q3)	Median (Q1–Q3)
C (n = 5)	39 (24–53)	4 (2–6)
IM1 (n = 5)	27 (23–41)	3 (3–5)
IM2 (n = 5)	24 (24–47)	2 (2–5)
IM3 (n = 5)	37 (32–44)	4 (3–5)
*p* value	0.538	0.582

**Table 2 vetsci-10-00600-t002:** Morphometric parameters of the reproductive tract.

Parameter	C (n = 5)	IM1 (n = 5)	IM2 (n = 5)	IM3 (n = 5)	SEM	*p* Value
Total genital tract weight (kg)	1.403 ^a^	0.508 ^b^	0.590 ^ab^	0.599 ^ab^	0.10	0.024
Segment cervix–vagina (cm)	40.60 ^a^	29.20 ^bc^	37.60 ^ab^	24.80 ^c^	1.82	0.001
Uterine corpus (cm)	11.20	14.40	11.00	13.00	0.90	0.523
Uterine horns						
Length (cm)	120.65	75.90	80.80	107.30	7.25	0.091
Diameter (cm)	4.95 ^a^	2.85 ^ab^	2.55 ^b^	2.50 ^b^	0.26	0.005
Ovaries						
Length (cm)	4.22 ^a^	2.99 ^b^	2.75 ^b^	3.25 ^ab^	0.18	0.040
Width (cm)	2.82 ^a^	2.27 ^b^	1.85 ^b^	2.45 ^ab^	0.13	0.039
Depth (cm)	2.00 ^a^	1.30 ^b^	1.10 ^b^	1.37 ^ab^	0.12	0.020
Volume (ml)	13.25 ^a^	4.72 ^b^	2.91 ^b^	6.58 ^ab^	1.29	0.015
Weight (g)	12.00 ^a^	3.50 ^b^	4.20 ^ab^	5.10 ^ab^	0.94	0.027
GSI	5.33 ^a^	1.57 ^b^	2.09 ^ab^	2.31 ^ab^	0.40	0.014

All values are means. C: control females; IM1: immunocastrated two weeks after farrowing; IM2: immunocastrated at the beginning of estrus; IM3: immunocastrated one week after the beginning of estrus; SEM: standard error of the mean. Different superscript letters (a, b, c) on the same line indicate significant differences.

**Table 3 vetsci-10-00600-t003:** Physicochemical parameters and boar taint compound quantification.

Parameter	C (n = 5)	IM1 (n = 5)	IM2 (n = 5)	IM3 (n = 5)	SEM	*p* Value
Live weight (kg)	219.36	223.18	209.96	222.40	7.47	0.933
Carcass weight (kg)	160.40	167.00	158.50	158.60	6.56	0.970
Carcass yield (%)	73.20	74.20	75.90	70.92	1.25	0.512
pH_45min_	6.45	6.14	6.27	6.30	0.10	0.846
pH_24h_	5.50	5.67	5.58	5.62	0.03	0.183
*L**	48.2	47.2	45.1	48.1	0.80	0.542
*a**	21.7	21.1	22.5	22.3	0.29	0.335
*b**	6.8	5.8	6.5	6.6	0.34	0.791
C*	22.7	21.9	23.4	23.2	0.35	0.487
h°	17.3	15.3	15.9	16.4	0.65	0.768
Heme (mg/g)	1.92	1.78	2.30	1.89	0.08	0.105
Drip loss (%)	3.25	2.33	3.22	3.40	0.23	0.206
Cooking loss (%)	23.31	23.73	21.06	24.09	0.61	0.299
Shear force (N/cm^2^)	56.89	64.46	81.63	69.81	4.68	0.311
Moisture (%)	71.09 ^b^	72.78 ^a^	71.19^b^	73.16 ^a^	0.27	0.001
Protein (%)	23.18	23.08	24.29	23.21	0.24	0.206
Intramuscular fat (%)	2.89	2.40	2.26	2.02	0.24	0.652
Ashes (%)	1.13	1.11	1.08	1.10	0.01	0.655
Androstenone (ng/g)	8.83	4.36	12.02	4.29	1.77	0.441
Skatole (ng/g)	4.07	7.25	3.91	3.51	0.56	0.109

All values are means. C: control females; IM1: immunocastrated two weeks after farrowing; IM2: immunocastrated at the beginning of estrus; IM3: immunocastrated one week after the beginning of estrus; SEM: standard error of the mean; *L**: lightness; *a**: redness; *b**: yellowness; C*: chroma; h°: hue. Different superscript letters (a,b) on the same line indicate significant differences (*p* < 0.05).

**Table 4 vetsci-10-00600-t004:** Fatty acid profile (g/100 g of fat) of LTL muscle.

Fatty Acid (g/100 g Fat)	C (n = 5)	IM1 (n = 5)	IM2 (n = 5)	IM3 (n = 5)	SEM	*p* Value
Total saturated fatty acid (SFA)	30.24	32.92	29.86	32.46	0.58	0.206
Myristic acid (C14:0)	1.04	1.12	1.00	1.10	0.03	0.648
Pentadecanoic acid (C15:0)	0.02	0.02	0.02	0.02	0.00	0.299
Palmitic acid (C16:0)	20.28	21.65	19.87	21.14	0.36	0.440
Margaric acid (C17:0)	0.11	0.11	0.12	0.13	0.00	0.309
Stearic acid (C18:0)	8.44	9.60	8.47	9.67	0.21	0.059
Arachidic acid (C20:0)	0.20	0.24	0.23	0.24	0.01	0.309
Total monounsaturated fatty acid (MUFA)	49.65	51.52	49.03	48.18	0.80	0.521
Palmitoleic acid (C16:1 n-7)	3.62	3.64	3.42	3.46	0.12	0.851
Oleic acid (C18:1 n-9)	38.61	40.48	38.09	37.51	0.72	0.444
Gondoic acid (C20:1 n-9)	0.84	0.92	0.87	0.88	0.03	0.550
Total polyunsaturated fatty acid (PUFA)	10.85	10.99	11.76	11.73	0.55	0.690
Linoleic acid (C18:2 n-6)	7.18	7.71	7.88	8.38	0.31	0.727
Alpha-linolenic acid (C18:3 n-3)	0.18	0.22	0.20	0.22	0.01	0.205
Dihomoγ-linolenic acid (C20:3 n-6)	0.19	0.21	0.23	0.21	0.01	0.374
Arachidonic acid (C20:4 n-6)	2.52	2.15	2.63	2.25	0.21	0.771
Eicosatrienoic acid (C20:3 n-3)	0.05	0.05	0.05	0.05	0.00	0.769
Eicosapentaenoic acid (C20:5 n-3)	0.05	0.05	0.06	0.04	0.00	0.452
Docosapentaenoic acid (C22:5 n-3)	0.31	0.23	0.30	0.22	0.03	0.690
Docosaheptaenoic acid (C22:6 n-3)	0.07 ^ab^	0.04 ^ab^	0.09 ^a^	0.02 ^b^	0.01	0.015
PUFA/SFA	0.37	0.34	0.39	0.37	0.02	0.612
Total n-3	0.65	0.59	0.70	0.55	0.04	0.666
Total n-6	10.20	10.40	11.06	11.18	0.51	0.684
n-6/n-3	15.74 ^b^	17.78 ^ab^	16.04 ^b^	20.42 ^a^	0.51	0.005

All values are means. C: control females; IM1: immunocastrated two weeks after farrowing; IM2: immunocastrated at the beginning of estrus; IM3: immunocastrated one week after the beginning of estrus; SEM: standard error of the mean. Different superscript letters (a, b) on the same line indicate significant differences.

## Data Availability

The data are available from the corresponding author upon reasonable request.

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
