# Peer review of "Effect of Immunocastration on Culled Sows—A Preliminary Study on Reproductive Tract, Carcass Traits, and Meat Quality"

_vetsci, 2023, doi:10.3390/vetsci10100600_

Round 1
Reviewer 1 Report
The manuscript entitled “Effect of immunocastration on culled sows – a preliminary study on reproductive tract, carcass traits and meat quality “ evaluated the effect of immunocastration on the reproductive tract as well as on the carcass and meat quality. This topic is of major interest for swine industry. The authors have done an interesting work. However, before considering publication of this paper, there are some major concerns that need to be cleared out.
Major comments & concerns
1. First and foremost, the material and method section is lacking of very important information. There is no information about the inclusion &exclusion criteria and the sample size calculation and power of this study is missing. Furthermore, several definitions of parameters should be presented in more detail.
2. In the introduction section current and relevant publications in this research area are missing.
3. The discussion is lacking of the limitations of the study.
I think these are important points that need to be addressed before this paper can be considered for publication in Veterinary Science. In addition, I have added some specific points and comments to the manuscript.
Abstract
Line 18: Please only use reproductive cycle and delete productive
Line 32: Please be consistent with sow taint or always use boar taint.
Line 34-35: Please present the significant parameters in detail. So the reader can see the data directly in the abstract.
Introduction
Line 46: Please add a current reference that describe the findings and delete the two other ones:
Diagnosis of endometritis and cystitis in sows: use of biomarkers; General and comparative aspects of endometritis in domestic species: A review.
Line 50: Please add this reference Management practices to optimize the parturition process in the hyperprolific sow; Lactation in swine: review article.
Line 80-85: Please add the countries of the presented data.
Material and Methods:
Line 132: Please add increased piglet mortality rate due to sow behaviour
Line 149: I suggest to write until the end of lactation instead of farrowing
Line 150: Please be more precise. At what day after weaning was the injection applied?
Line 152: Group 3 instead of 2
Line 177-178: Please describe in more detail how you measured length and width and depth in the different parts of the reproductive tract. Furthermore, describe how you differentiated the different parts of the uterus.
Did you also evaluate the mucosa of the reproductive tract and some pathological findings such as endometritis, pyometra, ovarian cysts? This can also influence the weight and length.
Have you evaluated the cycle stage of the ovarian and evaluated functional bodies?
Please describe how you calculated the volume of the ovary.
Line 275: Please describe which animals you used for sensory evaluation? Was this just a test run for development of the method?
Line 318-319: Please complete the sentence.
Please describe why you didn’t analyse indol for sow taint?
Results:
Line 326-327: Please give details of all animals for the culling reason. Since the animal number is very low this information is important for the study and might influence the reproductive tract.
Line 345-346: Please do not interpret the data in the result section. This should be discussed and concluded in the discussion. Therefore, remove the sentence.
Line 354: Be consistent with sow taint or boar taint.
Line 363-364: Please describe in the discussion section, why androstenone is higher in group 2 compared to the other groups.
Conclusion:
Please remove histological investigations in this section, because you already have proven that the lack of GnRH influenced the reproductive tract.
Minor editing of English language required
Author Response
We sincerely appreciate the valuable time and effort you dedicated to reviewing this manuscript. Below, you will find detailed responses to your feedback, along with the corresponding revisions and corrections, which have been highlighted or tracked in the resubmitted files.
Major comments and concerns
- This study was conducted under real farm conditions, not in experimental farms. Therefore, it has used a convenience population of animals, selected for culling due to unfitness regarding the farm's production goals; the farm was limited to 60 breeding sows. The study spanned from November 2019 to September 2020, as we had to wait for the animals to become available for the designated treatments. Given the real-world context, no sample size calculation or statistical power analysis was applicable to this study. We included this information in the manuscript.
- Current and relevant publications have been added as suggested.
- The limitations of the study have been added to the discussion.
Specific points and comments to the manuscript:
Line 18: ‘Productive’ word was deleted.
Line 32: The ‘boar taint’ term was kept in all the document.
Line 34-35: Detailed results were added in the abstract.
Line 46: The recommended references were added.
Line 50: The recommended references were added.
Line 80-85: The data presented there, respects the production of Bísaro pig, which is only found in the north of Portugal (DOP products with local production certification).
Line 132: The suggestion was accepted.
Line 149: The suggestion was accepted.
Line 150: The IM2 group, constituted of multiparous sows, was inoculated at the beginning of estrus. The number of days after weaning was not accounted for. Instead, the detection of heat was performed, using the standing reflex to define the onset of estrus. This information was introduced in the manuscript.
Line 152: Changed accordingly.
Line 177-178: The measurements and how we differentiate the different segments of the reproductive tract were detailed in the manuscript, as suggested. A new image (Figure 2) was introduced to illustrate the anatomical limits for the measurements of the reproductive tract measurements.
While we did identify some morphological alterations in the reproductive tract, they were not the primary focus of this manuscript, and we did not detail them in the text. In some cases, the uterus showed a limited amount of purulent or serous material, and the uterine condition was scored as a mild inflammatory condition based on the macroscopic appearance. Still, the liquid content was drained before weighting the genital tract. We also evaluated the functional aspects of the ovaries in relation to the estrous cycle (with the presence of follicles and corpus luteum), but as previously mentioned, this will be the subject of another manuscript.
Calculation of the ovary’s volume was done in accordance with Leonhardt et al., 2014 formula for ellipsoid structures (ovarian length x width x depth x 0.523). This sentence was added to the manuscript.
Line 275: All the studied animals were used for the sensory evaluation triangle tests (this sentence was added to the manuscript). Given the lack of significant differences observed in the analysis of boar taint compounds, it was anticipated that panelists would likewise struggle to discern distinctions among the samples. Consequently, the use of the triangular test was considered appropriate in this context, and our expectations were confirmed when no discernible differences emerged between the control groups and immunocastration treatments, in accordance with previous studies by our expert workgroup (references 53 and 54).
Line 318-319: The sentence was reworded for clarification.
Please describe why you didn’t analyse indol for sow taint? Androstenone and skatole are widely considered to be the main contributors to boar taint, as confirmed by numerous studies. However, animal tissues contain varying levels of other compounds, such as indole and other steroids, which could affect the perception of the main contributors to boar taint (Annor-Frempong et al., 1997a; Morlein et al., 2016). Verbatim excerpt from Meinert et al. (2017): However, there is a general consensus that two compounds are primarily responsible for boar taint: skatole (3-Methyl-Indole) and androstenone (5α-androst-16-en-3-one) (Patterson, 1968; Vold, 1970; Bonneau, 1982; Bonneau & Squires, 2004; Byrne, Thamsborg, & Hansen, 2008), even though other compounds have also been proposed, such as p-cresol and 4-ethylphenol (Patterson, 1967) and indole and androstanol (Fischer et al., 2011). Current research is therefore focused on skatole and androstenone.
Annor-Frempong, I. E., G. R. Nute, F. W. Whittington, and J. D. Wood. 1997a. The problem of taint in pork .1. Detection thresholds and odour profiles of androstenone and skatole in a model system. Meat Science 46(1): 45-55. https://doi.org/10.1016/s0309-1740(97)00003-x
Meinert, L., Lund, B., Bejerholm, C., and Aaslyng, M. D. 2017. Distribution of skatole and androstenone in the pig carcass correlated to sensory characteristics. Meat Science 127: 51-56. https://doi.org/10.1016/j.meatsci.2017.01.010
Morlein, D., J. Trautmann, J. Gertheiss, L. Meier-Dinkel, J. Fischer, H. J. Eynck, L. Heres, C. Looft, and E. Tholen. 2016. Interaction of Skatole and Androstenone in the Olfactory Perception of Boar Taint. Journal of Agricultural and Food Chemistry 64(22):4556-4565. https://doi.org/10.1021/acs.jafc.6b00355
Line 326-327: The first paragraph of the results discusses the reasons for culling, including factors like low prolificity, infertility, and various health-related issues. However, some important individual reproductive and clinical data were amiss, therefore we believe the data presented here is appropriate to our findings. Nonetheless, the veterinary practitioner evaluated the animals and found them healthy and suitable for transportation, according to the international laws. Despite the little information available in the farm clinical records, we are open to adding them if you deem essential.
Line 345-346: This sentence was removed.
Line 354: The term ‘sow taint’ was changed to ‘boar taint’.
Line 363-364: While group 2 exhibits numerically higher androstenone levels, no statistically significant differences exist between the groups. It is important to state that the observed values fall below the sensory threshold (residual values), rendering them virtually undetectable in the sensory evaluation. In addition, any putative effects related to the reproductive stage at the beginning of treatments seem less plausible, due to the time lapse that occurred between the first Improvac® injection and the slaughter or the androstenone determination. Therefore, the presence of these slightly elevated values may be attributed to genetic variability, leading certain sows to more efficiently convert sex steroids into androstadienone, and subsequently into androstenone, compared to their counterparts.
Conclusion: The sentence related to histological investigations was removed from the conclusion.
Reviewer 2 Report
General Comments
Immunocastration could be a solution to avoid undesirable pregnancies and sow taint in cull sows. This study aimed to investigate the effects of three different immunocastration protocols on culled sows. The authors showed that immunocastration can effectively suppress reproductive activity in culling sows, and had no impact on pork quality.
In general, this is an interesting study and can provide the related data for immunocastration performed in culling sows. However, it still has some concerns needs to be elucidated.
Specific Comments:
1. From the conclusion of the paper, immunocastration can effectively suppress reproductive activity in culling sows without negatively impacting pork quality, which means that immunocastration does not increase in their economic value on culled sows but lead to increased cost, as most sows are culled immediately after weaning their last litter. Does this mean that immunocastration should be abandoned in culled sows?
2. Three immunocastration protocols represent three different points, these is no need to emphasize different points in Simple Summary.
3. For all the tables, just one “SEM” and “p value” was presented for four group, please confirm the data.
4. “The immunocastration triggered a reduction in the size and weight of the reproductive tract in all treated groups compared with intact females”, this in not true such as longer size of Uterine Corpus in group IM2.
5. There is no difference in weight of genital tract between the group C and IM2, the group C and IM3, and no difference in weight of Ovaries between the group C and IM2, the group C and IM3. It is not true in line 340.
6. Morphometric differences were observed in Table 2 in different immunocastration protocols rather than no marked Morphometric differences were recorded.
7. Results of ovaries size were missing in the main text.
Minor editing of English language is required
Author Response
We sincerely appreciate the valuable time and effort you dedicated to reviewing this manuscript. Below, you will find detailed responses to your feedback.
- While it is true that most industrial sows are typically culled after weaning their last litter, this is not the usual practice for Bísaro pigs. This autochthonous breed experiences quite unique breeding conditions. Following the weaning of their last litter, sows are retained on the farm to improve their body condition before slaughter. During this period, they are kept in outdoor pens, often located near forests or agricultural fields, which poses a risk of wild boars breaching their enclosures and mating with them. In some smaller, family-run farms, sows intended for culling are also housed in outdoor pens adjacent to male pigs, which can result in the farm's own males mating with the females. Immunocastration offers an effective solution to address this issue, enabling the females to enhance their body condition before slaughter and ultimately raising their economic value. We included this information in the manuscript.
- The sentence was corrected in the manuscript, as suggested.
- The SEM presented is from all groups combined. Some authors used this approach, which is why we also presented our data with SEM from all the combined groups. Examples:
Aluwé, M., et al. "Chicory fructans in pig diet reduce skatole in back fat of entire male pigs." Research in Veterinary Science 115 (2017): 340-344.
Aluwé, Marijke, et al. "Effect of surgical castration, immunocastration and chicory-diet on the meat quality and palatability of boars." Meat Science 94.3 (2013): 402-407.
Hansen, Laurits Lydehøj, et al. "Effect of feeding fermentable fibre-rich feedstuffs on meat quality with emphasis on chemical and sensory boar taint in entire male and female pigs." Meat Science 80.4 (2008): 1165-1173.
Pauly, C., et al. "Growth performance, carcass characteristics and meat quality of group-penned surgically castrated, immunocastrated (Improvac®) and entire male pigs and individually penned entire male pigs." Animal 3.7 (2009): 1057-1066.
We chose this presentation method because it aligns with the approach used by other authors, but we are open to making changes if you believe a different approach would be more suitable.
- The sentence was corrected in the manuscript, as suggested.
- The sentence was corrected in the manuscript, as suggested.
- The sentence was corrected in the manuscript, as suggested.
- The ovaries size results were added to the main text, thank you so much for this input.
Once again, we are highly appreciative of the time and effort you put in improving our manuscript.
Reviewer 3 Report
The main issue in this study was to evaluate the effect of immunocastration of the Bísaro culled sows on the morphometry of the female reproductive tract, the skatole and androstenone levels, the carcass traits and meat quality, as well as on meat sensory evaluation, and comparison of the results with data obtained from uncastrated animals (entire females). An attempt was made to check whether different times of immunological castration (lactation, estrus, diestrus) are sufficient to ensure good preservation of most meat parameters.
1. I believe that the subject of this work is original and appropriate in this field, because in primitive and conservative (autochthonous) breeds (eg. Bísaro pigs) it is not always possible to successfully repeat the actions (immunocastration) tested in modern pig breeds.
In the present study, the authors for the first time, used the multiparous females with fully functional reproductive tracts. Compared to other studies on immunocastration in Bisaro pigs, this paper is also the first to discuss the assessment of most physicochemical and organoleptic parameters of carcasses of immunocastrated Bisaro sows. In the properly carried out and correctly described methodology of this work, the authors designed an appropriate number of animals and a large number of modern and traditional research methods. The authors obtained the results, thanks to which they could open a very interesting discussion, bringing a lot of new and interesting information closely related to the issues raised by them. The authors demonstrated high effectiveness of immunocastration in multiparous sows during lactation, estrus and in the period just after estrus.
1. A serious limitation of the work is that it was not shown in what period of the year the research was conducted (summer, spring, winter)?. Were the animals all on the farm together or on each farm separately during the research?
I noticed a lot of errors in the marking of the literature:
repeated publication, No. 7 and 18, the same
line 152: repetition of experimental group No. 2, no group No. 3?
line 285: publication no. 46 or 47?
line 291: publication no. 50 or 51?
line 291 – 293: the cited publications are probably not related to the text: (Sodring? Line 635-636-637)
line 306: publication no. 52 or 53?
Line 310: publication 52 or 53?
line 424: is 56 should be 57
line 427: is 58 should be 59
line 436: is 68 should be 69?
line 437: is 58, should be 59 (unclear repetition of text?)
line 457: 72 or 73?
Line 464: is 62 , should be 63!
line 480: is 80 should be 81
line 483: is 56 should be 57
line 483: is 59 should be 60
line 558: is animal should be: Animal
1. The conclusions are consistent with the evidence, and the arguments presented well answer the main question posed. The references presented are adequate, current and well chosen. The 4 tables and 1 figure raise no objections.
1
Author Response
We would like to express our sincere appreciation to the reviewer for dedicating his/her time and expertise to thoroughly assess our manuscript. Your invaluable insights and constructive comments have greatly contributed to the enhancement of our work.
The research was conducted from November 2019 to September 2020 due to the limited availability of test subjects. Specifically, both C and IM2 were carried out in the winter, IM1 in the spring, and IM3 in the summer. Despite recognizing the potential impact of seasonal variations on boar taint compounds, our study fails detect any significant differences. Throughout the research period, all animals remained on the farm and were grouped in outdoor pens to facilitate observation.
I apologize for the errors in the literature; they have all been rectified.
Round 2
Reviewer 1 Report
The authors significantly improved the manuscript and therefore, it can be published in the current form.